# Management of Patients Diagnosed with Endometrial Cancer: Comparison of Guidelines

**DOI:** 10.3390/cancers15041091

**Published:** 2023-02-08

**Authors:** Stefano Restaino, Chiara Paglietti, Martina Arcieri, Anna Biasioli, Monica Della Martina, Laura Mariuzzi, Claudia Andreetta, Francesca Titone, Giorgio Bogani, Diego Raimondo, Federica Perelli, Alessandro Buda, Marco Petrillo, Pantaleo Greco, Alfredo Ercoli, Francesco Fanfani, Giovanni Scambia, Lorenza Driul, Giuseppe Vizzielli

**Affiliations:** 1Clinic of Obstetrics and Gynecology, “S. Maria della Misericordia” University Hospital, Azienda Sanitaria Universitaria Friuli Centrale (ASUFC), 33100 Udine, Italy; 2Medical Area Department (DAME), University of Udine, 33100 Udine, Italy; 3Department of Biomedical, Dental, Morphological and Functional Imaging Science, University of Messina, 98125 Messina, Italy; 4Medical Area Department (DAME), Institute of Pathological Anatomy, Chief School of Specialization in Pathological Anatomy, “S. Maria della Misericordia” University Hospital, Azienda Sanitaria Universitaria Friuli Centrale (ASUFC), 33100 Udine, Italy; 5Department of Medical Oncology, “S. Maria della Misericordia” University Hospital, Azienda Sanitaria Universitaria Friuli Centrale (ASUFC), 33100 Udine, Italy; 6Department of Radiation Oncology, S. Maria della Misericordia” University Hospital, Azienda Sanitaria Universitaria Friuli Centrale (ASUFC), 33100 Udine, Italy; 7Department of Maternal and Child Health and Urological Sciences, Policlinico Umberto I, Sapienza University of Rome, 00185 Rome, Italy; 8Division of Gynaecology and Human Reproduction Physiopathology, IRCCS Azienda Ospedaliero-Univeristaria di Bologna, 40138 Bologna, Italy; 9Division of Gynaecology and Obstetrics, Santa Maria Annunziata Hospital, USL Toscana Centro, 50012 Florence, Italy; 10Gynecology Oncology Surgical Unit, Department of Obstetrics and Gynecology, Ospedale Michele e Pietro Ferrero, 12060 Verduno, Italy; 11Gynecologic and Obstetric Clinic, Department of Medicine, Surgery and Pharmacy, University of Sassari, 07100 Sassari, Italy; 12Dipartimento di Scienze Mediche, Università degli Studi di Ferrara, 44011 Ferrara, Italy; 13Department of Human Pathology in Adult and Childhood “G. Barresi”, Unit of Gynecology and Obstetrics, University of Messina, 98125 Messina, Italy; 14Dipartimento per le Scienze Della Salute Della Donna, del Bambino e di Sanità Pubblica, UOC Ginecologia Oncologica, Fondazione Policlinico Universitario Agostino Gemelli IRCCS, 00168 Rome, Italy

**Keywords:** endometrial cancer, management, molecular biology, guidelines

## Abstract

**Simple Summary:**

Endometrial cancer has a high epidemiological impact, and its management is part of everyday clinical practice. International guidelines have been arranged over the years according to major recent discoveries. The application of the guidelines released by different international gynecological societies is still matter of debate as they diverge in many issues. Authors wanted to compare them and point out the differences, aiming to both draw the attention to a need of unification and to provide a useful tool for clinicians.

**Abstract:**

Endometrial cancer is the most common gynecological malignancy in Europe and its management involves a variety of health professionals. In recent years, big discoveries were made concerning the management of patients diagnosed with endometrial cancer, particularly in the field of molecular biology and minimally invasive surgery. This requires the continuous updating of guidelines and protocols over the years. In this paper, we aim to summarize and compare common points and disparities among protocols for management of patients diagnosed with endometrial cancer by leading international gynecological oncological societies. We therefore systematically report the parallel among the guidelines based on the various steps patients with endometrial cancer usually undergo. The comparison between American and European protocols revealed some relevant disparities, in particular regarding surgical staging, molecular biology application as a prognostic tool and follow up regimens. This could possibly cause differences in interpreting and applying protocols in clinical practice in small centers, leading to a lack of adherence to guidelines or even prompting a confusing mix of them.

## 1. Introduction

Endometrial carcinoma is the most common gynecological cancer, accounting for 417.367 new cases worldwide in 2020 and nearly 100.000 deaths. The incidence has been rising over the years with aging and increased obesity of the high-income countries’ populations [1]. As it is mostly discovered in the early stages, surgery is often the only required treatment and patients receive close follow-up for many years.

The management of women who suffer with this kind of neoplasm involves a variety of health professionals including gynecological oncologists, general gynecologists, general surgeons, radiation oncologists, pathologists, medical oncologists, radiologists, general practitioners, and palliative care teams.

Over the years, international societies have developed guidelines to minimize variations in clinical practice and to improve patient outcomes. A consensus conference was held in 2014 in Europe to create multi-disciplinary evidence-based guidelines about specific topics on endometrial carcinoma [2]. In 2020, ESGO, ESTRO, and ESP in collaboration decided to update and extend these guidelines, enriching them with new discoveries in surgical procedures and molecular biology [3]. In addition, at the beginning of 2022, ESMO alone published its new clinical practice guidelines on endometrial cancer diagnosis, treatment, and follow-up [4]. The American College of Obstetricians and Gynecologists in April 2015 published its practice bulletin (No. 149), as a review of the actual knowledge and an integration of additional guidelines [5]. Many other national societies edited or updated their own guidelines. Here are some of the latest guidelines in Europe: the British Gynaecological Cancer Society issued the first endometrial cancer guidelines as a recommendation for practice for the UK in 2017; in the same year, the French society of gynecologic oncology (SFOG) and French college of obstetricians and gynecologists (CNGOF) produced their first joint recommendations; in 2021, SEOM (Spanish Society of Medical Oncology) provided the latest Spanish guidelines; and in 2022, AIOM (Italian Society of Medical Oncology), in collaboration with the Italian Society of Gynecology and Obstetrics, updated their directives [6,7,8,9]. Besides these European recommendations, NCCN updated its previous clinical practice guidelines on uterine neoplasms in November 2021 [10].

The purpose of this paper is to summarize and compare the common points and disparities among guidelines for the management of patients diagnosed with endometrial cancer by leading international societies. Given the importance in clinical management acquired by the molecular biology classification of endometrial cancer, we decided to include in this review only the guidelines updated with these issues, as they are now essential to provide the best standard of care.

## 2. Materials and Methods

This is a descriptive comparative review. We searched on PubMed and all websites of the world’s major gynecologic oncology societies for any publications about endometrial cancer management. American NCCN guidelines (2021) and European ESGO, ESTRO, ESP (2020), and ESMO (2022) guidelines met the inclusion criteria, as they are the most recent and they are updated with concerns regarding molecular classification. These guidelines summarize American and European points of view in the management of patients with endometrial cancer. We therefore compared them and derived disparities in clinical practice that could possibly be a basis for further investigation to eventually equalize differences. A particular focus is placed on molecular classification as a prognostic risk factor that could improve pre and post operatory risk stratification.

## 3. Results

We systematically report below the comparisons among the guidelines based on the various steps patients with endometrial cancer usually undergo, eventually summarizing the results with an overview table.

### 3.1. Pre-Operative Work-Up

Nearly 90% of women with endometrial carcinomas experience abnormal vaginal bleeding, most commonly in the postmenopausal period. Many physicians believe that endometrial cancer is a more treatable malignancy because the bleeding often urges patients to seek care when the disease is still at an early stage. However, evidence shows that the mortality rate has increased more rapidly than the incidence rate over the years [11]. To improve the outcome, physicians should closely identify and select high-risk patients to tailor treatment properly and provide the best long-term survival. All guidelines agree on the recommendation that all patients with endometrial cancer must be referred to a tertiary care center.

#### 3.1.1. Endometrial Sampling

Initial evaluation for known or suspected endometrial cancer includes the record of an accurate medical history and a physical and gynecological examination with transvaginal ultrasound. A histologic sampling, which can easily be obtained by office endometrial biopsy (with or without endocervical curettage), is needed to provide diagnosis. The NCCN guidelines suggest that endometrial biopsy should be sufficient for planning definitive treatment. However, office endometrial biopsies have a false-negative rate of about 10%. Thus, they recommend a negative endometrial biopsy in a symptomatic patient to be followed by a fractional dilation and curettage (D&C) under anesthesia [12]. Hysteroscopy, as well, could be useful in evaluating the presence of endometrial lesions, such as polyps, and to guide their removal. ESMO guidelines do not take a position on which is the best type of endometrial sampling, stating that both biopsy and D&C are acceptable initial approaches to the histological diagnosis of EC. The ESGO–ESTRO–ESP panel, instead, does not express anything at all about the type of sampling.

#### 3.1.2. Imaging

Based on NCCN guidelines, preoperative workup includes a chest X-ray and possibly a pelvic MRI to establish the origin of the tumor and to assess the local extent. Other imaging tests (CT and/or PET/CT) may be used to evaluate for metastatic disease based on clinical symptoms, abnormal physical or laboratory findings, or in the case of abnormality seen on a chest X-ray. Chest/abdomen/pelvis CT should be preferred in case of high-grade/high risk carcinoma to evaluate metastatic disease. A total body PET scan is highly specific for the assessment of lymph node metastases, but it is not routinely performed [13]. In the case of clear cell/serous/dedifferentiated carcinomas and carcinosarcoma MRI or chest/abdominal/pelvic, CT should be always performed before surgery to assess the presence of extrauterine disease.

European guidelines report the use of magnetic resonance imaging as being quite useful, as it is highly specific in the assessment of deep myometrial invasion, cervical stromal involvement, and lymph node metastasis. Nevertheless, the diagnostic performance of transvaginal ultrasound (if performed by an expert sonographer) is rather similar to MRI in detecting myometrial invasion and cervical involvement, and so MRI could be avoided in the majority of cases [14,15,16,17].

### 3.2. Definition of Prognostic Risk Groups Integrating Molecular Markers

Both NCCN and European guidelines agree on the pivotal role of an expert pathologist in determining tumor type and grade (using a modified binary FIGO grading, which merges grade 1 and grade 2 endometrioid carcinomas as low-grade and grade 3 as high-grade) [18]. The molecular classification adds another layer of information to the conventional morphologic features and therefore could and should be integrated both in the pathologic report of the endometrial biopsy (if possible), and in the definitive surgical sample.

In fact, conventional pathologic analysis remains the standard for tumor risk stratification but suffers from inter-observer variation in establishing prognostic groups. The introduction of a diagnostic algorithm based on immunohistochemical and molecular markers, in order to apply a molecular classification and identify prognostic groups, is mandatory to reduce interobserver variation. Based on the results of The Cancer Genome Atlas, the ProMisE (Proactive Molecular Risk Classifier for endometrial cancer) algorithm proved to be a simple and accurate tool to overcome the lack of inter-observer diagnostic reproducibility and risk stratification by assigning patients with endometrial cancer to one of four groups based on a combination of mutation and protein expression analysis. Tumors are assessed for the presence of mismatch repair (MMR) proteins, for polymerase-epsilon (POLE) exonuclease domain mutations (EDMs) and for protein 53 (p53) immunohistochemistry, establishing four subgroups: MMR-D, POLEmut, p53 wild type (copy number low–CNL-or non-specific molecular profile-NSMP) or p53 null/missense mutations (copy number high) [19]. ESGO guidelines underline that in order to apply this molecular classification, all these diagnostic tests need to be performed, due to the non-negligible occurrence of double or multiple classifiers (almost 5% of tumors show a combination of positive tests). NCCN instead underlines that the decision to use molecular testing depends on the availability of the resources of each center but points out especially the importance to identify women with MMR deficiency. In fact, approximately 10% of MMR deficient/microsatellite unstable carcinomas are related to germline mutations of one of the MMR genes (MLH1, MSH2, MSH6, PMS2). So, this has not only a prognostic relevance, but permits to rapidly refer women to genetic counseling and possibly direct therapeutic decisions.

Molecular classification should guide clinical management when adjuvant chemotherapy is a conceivable option (high-grade/high-risk disease). In fact, validation of the molecular classification in high-grade and/or high-risk endometrial carcinomas shows that the POLE-mut tumors have an excellent prognosis, while the p53-abn neoplasms have a poor prognosis. On the other hand, in low-risk and intermediate-risk endometrial carcinoma with low-grade histology, POLE mutation analysis may be omitted as it will not change clinical practice. ESGO, ESTRO, and ESP guidelines recommend that if molecular classification tools are not available, endometrial carcinoma classification should be based on traditional pathologic features.

In conclusion, NCCN guidelines encourage molecular profiling (especially for research of MMR-d) to complement morphologic assessment of tumor type. However, only European guidelines propose a stratification of risk based on molecular findings.

### 3.3. Surgical Staging

The standard surgical procedure for patients with endometrial carcinoma is total hysterectomy with bilateral salpingo-oophorectomy (TH/BSO) and lymph node assessment (except in patient candidates for fertility-sparing options). The preservation of ovaries can be an option in premenopausal patients with FIGO stage IA G1 endometrial cancer. Ovarian preservation, though, is not recommended for patients with genetic risk for ovarian cancer (e.g., germline BRCA mutation, Lynch syndrome). The surgical approach could be laparoscopic, robotic, vaginal, or abdominal. However, the standard in those patients with apparent uterine-confined disease is to perform the procedure via a minimally invasive technique. This grants a lower rate of surgical site infection, hemorrhage, venous thromboembolism, a decreased hospitalization time, and a lower cost of care, without compromising oncologic outcome [20,21]. Both NCCN and European guidelines agree on a minimally invasive surgical approach, even in patients with high-risk endometrial carcinoma [22]. Moreover, both advise against intra-peritoneal tumor spillage, including tumor rupture or morcellation (including in a bag). If the risk of uterine rupture by vaginal extraction is high, other measures should be taken (e.g., mini laparotomy, use of endobag). Tumors with metastases outside the uterus and cervix (excluding lymph node metastases) are relative contraindications for minimally invasive surgery, but some selected metastatic patients with symptomatic endometrial carcinoma (e.g., relentless metrorrhagia) are also candidates for palliative hysterectomy using the less invasive procedure. In patients with serous endometrial carcinoma, carcinosarcoma, and undifferentiated carcinoma, due to a high risk of microscopic omental metastases, primary treatment includes total hysterectomy, bilateral salpingo-oophorectomy with surgical staging, peritoneal washing for cytology, omental sampling, peritoneal biopsies, and tumor debulking for gross disease. Even in these cases, minimally invasive surgery is the preferred technique if feasible. A major difference among guidelines is the type of omental sampling. NCCN suggests omental biopsy, while European guidelines advise infra-colic omentectomy. Moreover, NCCN guidelines consider omental biopsies even in clear cell carcinoma, while ESGO states not to perform omental sampling in this setting of patients, due to the low rate of omental metastasis in clear cell carcinomas.

In stage III and IV endometrial carcinoma (including carcinosarcoma), maximal cytoreduction should be considered only if macroscopic complete resection is feasible with acceptable morbidity (both guidelines agree). Even more, in these cases, surgery should be performed in a specialized center.

Positive peritoneal cytology is a negative prognostic factor, although it does not impact FIGO staging. Both guidelines agree on still obtaining cytology results because positive cytology may be useful to guide clinical decisions.

#### 3.3.1. Lymph Node Assessment

Lymph node assessment has still a pivotal role in surgical staging in patients with uterine-confined endometrial carcinoma, as it provides important prognostic information that may guide treatment decisions. The standard procedure comprises a pelvic nodal dissection (external iliac, internal iliac, obturator, and common iliac nodes), with or without para-aortic nodal dissection.

Since 2016, sentinel node biopsy has been introduced as an alternative to lymph node dissection for surgical staging. It is shown that it grants a similar prognostic accuracy, albeit permitting to a lower risk of post-operative morbidity, especially leg lymphedema [23]. If performed properly, a negative sentinel node confirms a pN0, with a high sensitivity in patients with early-stage endometrial carcinoma. This supports the impact of sentinel lymph node biopsy on establishing surgical management and potential adjuvant therapies [24,25,26,27,28,29]. Moreover, sentinel lymph node ultra-staging (that is the research of low volume metastasis by sectioning and staining all blocks of lymph node with hematoxylin–eosin, with or without cytokeratin immunohistochemistry) permits a more specific pathologic assessment. Specifically, it permits the detection of small metastases which could be missed by standard evaluation [30,31]. European guidelines state that sentinel lymph node mapping has to be preferred to systematic lymphadenectomy in patients with low-risk or intermediate risk carcinomas. The technique consists of the cervical injection of indocyanine green and the laparoscopic detection and removal of the green-colored lymph node on both pelvic sides. In case of non-detection on either pelvic side, the protocol requires a side specific lymphadenectomy, with pelvic and para-aortic infrarenal lymph node dissection. If pelvic lymph node involvement is found intra-operatively, a systematic pelvic lymph node dissection should be omitted, though the bulky lymph node should be removed. As a prognostic tool, the presence of both macro and micro metastases (<2 mm) should be considered as metastatic involvement, which is associated with a worse prognosis. In contrast, the prognostic significance of isolated tumor cells (ITC, pN0(i+), is still uncertain. There is no evidence that the presence of ITC has an impact on stage, but they should be taken into account when deciding on adjuvant therapy. In high-intermediate or high-risk disease, the ESGO panel suggests systematic lymphadenectomy, always including both pelvic and para-aortic node dissection. The NCCN panel equally defines sentinel lymph node sampling as the preferred technique for node evaluation if feasible in uterine confined disease. Otherwise, it recommends para-aortic systematic lymphadenectomy to be performed, only in patients with high-risk disease with deeply invasive lesions, high-grade histology, serous or clear cell carcinoma, or carcinosarcoma. SLN can be omitted (equally to systematic lymphadenectomy) in case no myometrial invasion is reported.

#### 3.3.2. Fertility-Sparing Therapy

Conservative management may be considered in highly selected patients with grade 1, stage IA endometrioid endometrial carcinoma who wish to preserve their fertility [32,33,34,35]. Both American and European guidelines advise for patients who desire a conservative treatment to be referred to specialized centers, and to receive counseling detailing that a fertility-sparing therapy is not the standard of care. TH/BSO with surgical staging is recommended after childbearing is complete, if therapy is not effective (endometrial cancer still present after 6 months of progestin-based therapy—6 to 12 months for NCCN) or if progression occurs. The ESGO panel suggests hysteroscopic biopsy, due to its higher agreement with the final diagnosis compared to dilatation and curettage, while NCCN does not express a preference. The ESGO underlines the need for these patients to be evaluated with an expert transvaginal ultrasound examination or with an MRI to exclude myometrial infiltration or cervical stromal invasion. Treatment consists of continuous progestin-based therapy and may include megestrol acetate (160–320 mg/day), medroxyprogesterone acetate (400–600 mg/day), or an intrauterine device containing levonorgestrel, better in association with oral progestins [36,37].

European guidelines suggest considering hysteroscopic resection prior to progestin therapy, as existing data suggest that patients who received hysteroscopic resection followed by progestin therapy achieve the highest complete remission rate [38,39,40].

To assess response, close follow up with endometrial sampling must be performed. The American panel recommends monitoring with either biopsies or dilation and curettage every 3 to 6 months. European guidelines instead advise for a closer surveillance with hysteroscopic guided biopsy at 3 and 6 months. If no response is achieved after 6 months, standard surgical treatment is recommended. The NCCN instead, in patients with persistent endometrial carcinoma after 6 months of failed hormonal therapy, recommends pelvic MRI to exclude myo-invasion and nodal/ovarian metastasis before continuing fertility-sparing therapy. After regression, the ESGO recommends strict follow-up with TVUS and physical examination every 6 months, with endometrial sampling to be collected just in case of abnormal uterine bleeding or atypical ultrasound findings.

### 3.4. Adjuvant Treatment

Adjuvant treatment mightily depends on the prognostic risk group. However, whilst ESGO–ESTRO–ESP and ESMO guidelines propose a different management according to the five risk categories (as reported in Table 1), the NCCN divides patients into those with a uterine confined disease, those with an extra-uterine disease, and those with a recurrent-metastatic disease. In fact, NCCN does not consider molecular biology as a tool for the definition of prognostic group and adjuvant therapy. In practice, a disease limited to the uterus could potentially be included in both low, intermediate, high-intermediate, and high-risk groups according to grade, lymph vascular space invasion and—most of all—molecular characterization.

Low risk endometrial cancer

Low-risk endometrial cancer includes (considering molecular classification) all stage I-II endometrial carcinomas POLE ultra-muted and all stage IA MMRd/NSMP endometrioid carcinomas, low grade, with lymph vascular space invasion (LVSI) negative or focal. For these patients, according to European guidelines, no adjuvant treatment is recommended [41,42,43].

Intermediate risk

Intermediate risk includes patients with stage IB MMRd/NSMP endometrioid carcinomas and low grade, no LVSI or focal invasion, patients with stage IA MMRd/NSMP endometrioid carcinomas and high grade, no LVSI or focal invasion and stage IA p53abn, and/or non-endometrioid carcinomas. In those patients, adjuvant brachytherapy can be recommended to decrease vaginal recurrence. It is shown to provide excellent control and high survival rates, similar to those after adjuvant external beam radiotherapy (EBRT) [44,45,46,47].

High–intermediate risk

This group includes stage IA or IB low grade endometrioid carcinomas with substantial LVSI or stage IB high-grade endometrioid carcinomas regardless of LVSI, and stage II endometrioid carcinomas. Considering the higher risk of recurrence (even with negative nodes), adjuvant radiotherapy is recommended, as it reduces the incidence of pelvic and para-aortic nodal relapse. Adjuvant vaginal brachytherapy alone could be considered in cases without substantial LVSI and for stage II low grade endometrioid carcinomas [48]. European guidelines suggest considering adding concomitant or sequential CHT to EBRT in cases of high grade and substantial LVSI. However, studies show controversial results in terms of recurrence-free survival and overall survival between adjuvant chemotherapy and EBRT alone. The preferred regimen in case of disease limited to the uterus is carboplatin/paclitaxel [49,50,51]. The NCCN instead stratifies patients with stage I endometrial cancer completely surgically staged by adverse risk factors: age (> than 60 years), LVSI positive, tumor size, lower uterine segment, or surface glandular involvement). In practice, in the case of stage IA low grade (G1 or G2), observation and close follow up are preferred (considering vaginal brachytherapy only if LVSI+ and age >/= than 60 years. In the case of stage IA high grade (G3), vaginal brachytherapy is the preferred option, considering eventually EBRT if LVSI+. In the case of stage IB, low grade vaginal brachytherapy is preferred (considering EBRT if G2 and LVSI+), while in case of stage IB high grade, the favorite regimen includes EBRT and/or vaginal brachytherapy with or without systemic therapy. The NCCN guidelines recommend EBRT with or without systemic therapy to all stage II (all grades).

High risk

According to European guidelines, high risk includes patients with stage III–IVA MMR-d or NSMP endometrioid carcinomas with no residual disease and stage I–IVA p53abn all-histology or stage IB-IVA non-endometrioid carcinomas without residual disease. In multiple studies, there is a reported benefit on survival rates in patients with advanced stage endometrial carcinoma treated with post-operative combined chemo-radiotherapy (delivered by either the sandwich or sequential method), compared with radiotherapy or chemotherapy alone [52,53,54,55,56,57,58,59]. With the best results in stage III carcinomas and in serous carcinomas regardless of stage, the PORTEC-3 trial showed better outcomes in terms of overall survival and failure-free survival benefit in patients who underwent combined chemotherapy and radiotherapy. The GOG-258 trial showed instead that combined chemotherapy-radiotherapy compared to chemotherapy alone grants significantly lower rates of pelvic and peri-aortic nodal relapse [51,60]. All this considered, EBRT with concurrent or sequential chemotherapy is recommended in these high-risk patients, with extended field RT in the case of involved para-aortic or common iliac nodes. An additional brachytherapy boost can be considered in the case of substantial LVSI, endocervical stromal invasion, and/or stage IIIB-IIIC. Chemotherapy alone is an alternative option. The NCCN recommends the use of a combination of chemotherapy and radiotherapy in this group, considering the high risk of recurrence. Radiotherapy alone is not recommended. Again, molecular classification is not mentioned. In the case of clear cell and serous endometrial carcinoma, the NCCN suggests the use of systemic therapy with or without EBRT in the case of both non-invasive stage IA with positive washings, both stages IB to IV, even if both guidelines agree that the definite benefit of added chemotherapy is unclear for patients with stage I–II clear cell carcinomas. Adjuvant platinum–taxane chemotherapy is the preferred regimen in patients with uterine serous carcinoma and clear cell carcinoma. European guidelines again suggest an approach based on molecular classification, as about 40–50% of clear cell and serous carcinomas are p53abn and must be included in high-risk carcinomas, with comparable outcomes. In fact, results of the PORTEC-3 trial showed a statistically significant survival advantage for p53abn carcinomas with combined therapy for stage I–III. In contrast, while POLE-ultra muted carcinomas had almost no recurrences even without adjuvant therapy, there was no clear benefit of added chemotherapy for MMRd and NSMP carcinomas [61]. In case of undifferentiated/dedifferentiated carcinoma or carcinosarcoma, adjuvant chemotherapy with concurrent EBRT/brachytherapy is always indicated and highly individualized. The NCCN panel now considers carboplatin/paclitaxel the preferred adjuvant therapy regimen for uterine-confined or recurrent/metastatic carcinosarcoma. A regimen based on the combination of ifosfamide and paclitaxel could be an option in this histology [62].

Advanced disease

The outcomes of advanced or recurrent disease remain unfavorable, with 5-year OS rates of 20–25% [63]. In stage III and IV endometrial carcinomas and in carcinosarcomas, upfront surgery with tumor debulking should be performed when a complete resection is feasible, still granting an acceptable morbidity and quality of life for the patient. State of art recommends, if surgery is not feasible, the use of primary systemic or local targeted therapy. For patients with unresectable locally advanced disease but no evidence of multiple distant metastases, treatment options include definitive radiotherapy (intended as EBRT followed by a boost of image guided brachytherapy) or neoadjuvant chemotherapy followed by surgery or definitive radiotherapy, depending on individual response. In these specific cases, ESGO guidelines suggest considering concurrent chemotherapy to enhance the radiation effect and adjuvant chemotherapy following primary local treatment (surgery or radiotherapy) to reduce the risk of distant metastases [64,65,66,67,68]. According to the ESGO panel, the treatment of residual lymph node disease after surgery should be based on a combination of chemotherapy and EBRT or chemotherapy alone to lower the risk of distant metastases. To reduce the risk of toxicity to surrounding tissue, an integrated or sequential boost in order to escalate the radiation dose, and an IMRT technique should be used [69]. Patients with residual pelvic disease after surgery (incomplete resection, positive margins, pelvic side wall or vaginal disease) seem to have a high risk of both local and distant recurrence. European guidelines in this setting of patients suggest an individualized approach with either radiotherapy or chemotherapy or a combination of both modalities. The table below briefly summarizes the comparison between American and European guidelines about adjuvant treatment (Table 2).

### 3.5. Management of Recurrences

Management of patients with a recurrence requires a multi-disciplinary approach. The choice among surgery, radiotherapy, and/or systemic therapy depends on tumor dissemination, type of prior treatment, interval between primary treatment and recurrence, and performance status and wishes of the patient [70]. Both European and NCCN guidelines divide recurrences into locoregional, oligometastatic/isolated metastasis or disseminated disease. A further distinction must be made among those who are radio naïve and those already treated with previous radiotherapy to the site of recurrence.

In the case of local or regional recurrences (vaginal cuff or pelvic tissues or pelvic-para-aortic lymph nodes):In the case of no prior radiotherapy exposure, both European and American guidelines recommend EBRT plus brachytherapy (1st choice), with or without subsequent chemotherapy. In the case of vaginal cuff recurrences, the ESGO–ESTRO–ESP panel suggests EBRT with or without brachytherapy (with brachytherapy alone suggested in the case of superficial tumors). It could be considered to surgically remove a solitary easily accessible superficial vaginal tumor prior to radiotherapy for better local symptom control [71,72,73,74].In the case of previous brachytherapy only, the NCCN recommends surgical exploration. If the disease is confined to the vagina or paravaginal soft tissue, EBRT plus brachytherapy is recommended. European guidelines, as well, advise EBRT with a brachytherapy boost. In the case of locoregional nodal disease, to the pelvic or para-aortic lymph node, both advise that EBRT with or without chemotherapy is the suggested approach. In the case of upper abdominal or peritoneal recurrence, systemic therapy is indicated with palliative radiotherapy if necessary.In the case of previous radiotherapy at the recurrence site, both the NCCN and ESGO–ESTRO–ESP guidelines suggest surgical exploration with radical resection as the preferred approach when feasible, followed by systemic therapy with or without radiotherapy. If surgery is not feasible, radical re-irradiation is the best option. The role of complementary chemotherapy after surgery for recurrence is not well established. Hence, the indication for chemotherapy should be evaluated on an individualized basis.

In the case of isolated distant metastases, the NCCN suggests considering surgical resection if feasible, or alternatively selected stereotactic radiotherapy. The ESGO–ESTRO–ESP and ESMO panels emphasize that surgery should be considered only when complete resection of macroscopic disease can be achieved with a reasonable morbidity profile. Even in the case of oligometastatic disease (with one to five metastasis sites), surgery if feasible is the best option. Otherwise, radiation therapy including stereotactic radiotherapy is the second choice. Systemic and/or radiation therapy should be considered post-operatively depending on the extent and pattern of relapse, and the amount of residual disease.

Systemic therapy is the first choice in the case of disseminated metastasis, patients non amenable to local treatment or further recurrences. In this regard, both the NCCN and European guidelines suggest hormonal therapy (in patients with low grade, asymptomatic, and hormone receptor-positive metastases), reserving chemotherapy for progression. Then, the ESGO guidelines recommended the use of medroxyprogesterone acetate and megestrol acetate; the alternative options include aromatase inhibitors, tamoxifen, fulvestrant. However, European guidelines underline that there are no universally agreed upon recommendations to predict a response to hormonal therapy in endometrial carcinoma based on hormonal receptor immunohistochemical status. The assessment of estrogen and progesterone receptor status in the primary tumor may not reflect the status in the recurrent or metastatic tumor, and thus a biopsy of recurrent or metastatic carcinomas for hormone receptor analysis may be helpful.

Chemotherapy is indicated in case of symptomatic, high grade, large volume metastases. Multiagent regimens are preferred, if tolerated. Recommended regimens include carboplatin–paclitaxel, cisplatin–doxorubicin, cisplatin–doxorubicin–paclitaxel, carboplatin–docetaxel, carboplatin–paclitaxel–bevacizumab, ifosfamide–paclitaxel or cisplatin–ifosfamide (for carcinosarcoma), carboplatin–paclitaxel–trastuzumab (for HER2-positive serous carcinoma), and everolimus–letrozole (for endometrioid histology). Compared to other regimens, carboplatin–paclitaxel is shown to have similar oncologic outcomes in terms of response rate and overall survival, with a more favorable toxicity and tolerability profile. Both NCCN and European guidelines advise the use of carboplatin–paclitaxel as first line [75,76,77]. For patients in whom paclitaxel is contraindicated, docetaxel can be considered in combination with carboplatin [78]. If multiagent chemotherapy regimens are contraindicated, single-agent therapy options can be used and include paclitaxel, cisplatin, carboplatin, doxorubicin and liposomal doxorubicin, topotecan, and docetaxel [79,80]. Responses range from 21% to 36%, apart from docetaxel, which is recommended for use as a single agent, but it is less active (7.7% response rate) than other agents [81].

No standard treatment has been identified as a second-line therapy, and a response rate of about 10–15% has been seen among all the available treatment options. According to the ESGO panel, doxorubicin and paclitaxel are considered the most active therapies, but more molecules are currently under study. In the case of recurrence of MSI-H/dMMR endometrial tumors, the NCCN and ESGO panels include pembrolizumab and dostralimab as treatment options. Moreover, based on recent clinical trials, the NCCN panel approves bevacizumab or temsirolimus as being appropriate for single-agent biologic therapy for patients who have progressed on previous cytotoxic chemotherapy [82,83]. Lastly, European guidelines propose a platinum-based chemotherapy re-challenge as an option for selected patients who relapse > 6 months following their last platinum-based therapy.

Clinical trials or best supportive care are appropriate for patients with disseminated metastatic recurrence who have a poor response to hormonal therapy and chemotherapy. The table below summarizes the comparison between guidelines about management of recurrences (Table 3).

### 3.6. Follow-Up

Patients diagnosed with endometrial cancer (all stages) have a 5-year survival rate of 84% and they often go through close follow-up for many years [84]. Surveillance protocols should be differentiated based on individual tumor risk, as patients with clinical stage I and II proved to have a significantly lower recurrence rate (15% in stage I-II vs. 50% in III–IV). Apart from tumor stage, there were varying definitions of risk across studies (including histology and histological grading) [85,86,87].

For most recurrent patients, disease relapse occurs within 3 years of initial treatment and most of them are symptomatic. Therefore, patients with vaginal, rectal, or urinary bleeding, decreased appetite and weight loss, abdominal pain, cough, and shortness of breath should seek prompt evaluation.

ESGO–ESTRO–ESP guidelines do not express this concern. According to ESMO guidelines, medical surveillance must be adjusted to risk factors. Therefore, for low-risk groups, they suggested frequency of follow-up every 6 months with physical and gynecological examination for the first 2 years, and then yearly until the 5-year mark. Moreover, in this group of patients, phone follow-up can be an alternative, after adequate patient education regarding concerning signs and symptoms of relapse [88]. In the high-risk groups (basically advanced disease, stage III–IV, high grade, non-endometrioid histology, node involvement), instead, physical, and gynecological examinations are recommended every 3 months for the first 3 years, and then every 6 months until the fifth year. Routine CT scans are not recommended, but they could be considered in the high-risk group, particularly if there was node extension (e.g., every 6 months for the first 3 years and then on an individual basis). PET-CT has been shown to be more sensitive and specific for the assessment of suspected recurrent EC, but its indication must be individualized. The accuracy of cancer antigen 125 is low and Pap smear test is not useful for detecting local recurrences, so they are both not routinely recommended during follow up.

The NCCN, while rejecting intensive surveillance, does not make a clear distinction between high and low risk, proposing an individualized scheme based on symptomatology and clinical issues. According to the American panel, a physical exam with transvaginal ultrasound should be performed every 3-6 months for 2 or 3 years, then every 6 months up to the fifth year, and then annually. Additional imaging (chest–abdomen–pelvis CT scan, PET-CT, or MRI) is helpful if clinically indicated, in the case of physical findings suspicious of recurrence or in the case of patients treated with a stage III–IV disease. In this last setting, chest–abdominal–pelvic CT is suggested as an optional recommendation every 6 months during the first 3 years of surveillance, and every 6 to 12 months for 2 additional years to early detect asymptomatic recurrences. CA 125 must be dosed regularly only if initially elevated, and the use of vaginal cytology is no longer recommended for asymptomatic patients [89,90]. Lastly, the NCCN promotes health maintenance as part of the follow-up schedule: blood pressure monitoring, breast examination, mammography as clinically indicated, stool test, immunizations. Patients should also receive counseling and education regarding lifestyle, obesity, exercise, smoking cessation, sexual health, nutrition, and potential late or long-term effects of treatment, and this has been shown to provide psychosocial reassurance and improve quality of life for patients and their families [91,92]. Table below summarizes the comparison of follow-up schemes (Table 4).

## 4. Discussion

Considering the big concern of new molecular findings, first, the substantial differences in implementation of molecular classification for risk stratification should be noted. Based on the huge number of recent studies, while American guidelines maintain a more conservative profile, the European panel reassessed the entire adjuvant therapy scheme based on the new classification. This allows for an even more individualized approach and permits avoiding unnecessary treatments with potential side effects in patients with a reassuring molecular profile. Studies to evaluate changes in prognostic risk profiles by comparing clinical risk assessment with the integrated molecular risk assessment profiling have already been published. It has been widely shown that molecular categorization of endometrial cancer allows the reallocation of a considerable proportion of patients in a more accurate prognostic group, and so decreases the use of adjuvant therapies to spare morbidity [93,94,95,96,97].

On the other hand, molecular testing permits the identification of women with MMR deficiency and thus enables their rapid referral to genetic counseling, which can possibly direct protective or therapeutic decisions. Moreover, knowing the molecular state of the cancer even before the surgical procedure or disease staging permits directing management to a more/less intensive scheme, for example offering a different preoperative workup. Further considerations must be made regarding the management of recurrences. European and American guidelines agree on proposing a scheme of treatment based on the evaluation of previous therapy and site and extent of recurrence. However, especially in this case, the approach needs to be individualized and the choice of the most adequate procedure is left to the multidisciplinary tumor board who has discussed and knows the patient. This results in a lower adherence to guidelines in general.

Regarding surgical staging, the guidelines show great differences in terms of omental sampling and systematic lymphadenectomy, as the European panel tends to be more radical (infra-colic omentectomy vs. omental biopsy, para-aortic lymphadenectomy routinely performed vs. only in high-risk patients).

Another big concern is post-treatment surveillance, as the follow-up is usually long and requires a considerable investment of clinical resources. Two main issues are disparities among follow-up schemes and a lack of adherence to the guidelines. In fact, in the absence of clear evidence from randomized studies, the intensity of follow-up regimens after surgical treatment of endometrial cancer is highly variable in clinical practice. It might be useful to make a clear and universally accepted distinction between low and high risk of relapse patients to precisely tailor follow-up regimes. This could help single centers to apply the protocol more evenly and closely. In addition, recent studies proved that there is a scarcity of evidence supporting the effectiveness of an intense follow up in improving survival or quality-of-life [98]. The results of the TOTEM study, a large randomized, multicenter, Italian study, showed no improvement in overall survival and early detection of relapses for patients followed with a 5-year intensive regimen, independently from their risk of relapse [99]. The TOTEM study proved that there is no reason to add routinely vaginal cytology, lab tests, or imaging investigations to the minimalist follow-up regimen, even in high-risk patients. The next step is to spread the news and to apply the minimalist regimen in clinical practice, daring to prescribe additional exams only if clinically indicated.

## 5. Conclusions

In conclusion, the comparison between American and European protocols revealed some relevant disparities in the management of patients diagnosed with endometrial cancer. This could possibly cause differences in interpreting and applying protocols in single centers, leading to a lack of adherence to guidelines, or even a mixing of them. Further efforts should be made to overcome these differences. Equalizing them might lead to a more homogeneous classification, surgical treatment, risk stratification, and follow up.

## Figures and Tables

**Table 1 cancers-15-01091-t001:** Definition of prognostic risk groups, Concin N, et al. Int J Gynecol Cancer 2020.

Risk Group	Molecular Classification Unknown	Molecular Classification Known
Low	Stage IA endometrioid + low-grade (G1-2) + LVSI negative or focal	Stage I–II POLEmut, no residual disease Stage IA MMRd/NSMP endometrioid + low-grade (G1-2) + LVSI negative or focal
Intermediate	Stage IB endometrioid + low-grade (G1-2) + LVSI negative or focal Stage IA endometrioid + high-grade (G3) + LVSI negative or focal Stage IA non-endometrioid (serous, clear cell, undifferentiated carcinoma, carcinosarcoma, mixed) without myometrial invasion.	Stage IB MMRd/NSMP endometrioid + low-grade (G1-2) + LVSI negative or focal Stage IA MMRd/NSMP endometrioid + high-grade (G3) + LVSI negative or focal Stage IA p53abn and/or non-endometrioid (serous, clear cell, undifferentiated carcinoma, carcinosarcoma, mixed) without myometrial invasion
High-intermediate	Stage I endometrioid + substantial LVSI regardless of grade and depth of invasion. Stage IB endometrioid high-grade G3 regardless of LVSI status Stage II	Stage I MMRd/NSMP endometrioid + substantial LVSI regardless of grade and depth of invasion Stage IB MMRd/NSMP endometrioid high-grade G3 regardless of LVSI status Stage II MMRd/NSMP endometrioid.
High	Stage III–IVA with no residual disease Stage I–IVA non-endometrioid (serous, clear cell, undifferentiated carcinoma, carcinosarcoma, mixed) with myometrial invasion, and with no residual disease.	Stage III–IVA MMRd/NSMP endometrioid, with no residual disease Stage I–IVA p53abn endometrial with myometrial invasion, with no residual disease Stage I–IVA NSMP/MMRd serous, undifferentiated carcinoma, carcinosarcoma with myometrial invasion, with no residual disease
Advanced disease	Stage III–IVA with residual disease Stage IVB	Stage III–IVA with residual disease of any molecular type Stage IVB of any molecular type

**Table 2 cancers-15-01091-t002:** Adjuvant treatment stratified on risk group.

Adjuvant Treatment	ESGO-ESTRO-ESP, ESMO	NCCN
Low	No adjuvant treatment recommended.	IA low grade (G1 or G2), no adjuvant treatment or vaginal brachytherapy if LVSI + and >/= 60 years IA high grade, vaginal brachytherapy + EBRT if LVSI+.
Intermediate	brachytherapy or EBRT	
High-intermediate	RT + CHT (in cases of high grade and substantial LVSI) or brachytherapy alone (if LVSI - and stage II low grade endometrioid)	
High	EBRT + CHT + brachytherapy boost if substantial LVSI, endocervical stromal invasion and/or stage IIIB-IIIC. or CHT All non-endometrioid carcinomas already included in high risk.	EBRT + CHT Clear cell/serous: stage IA with positive washings to stage IV, CHT +/− EBRT * Undifferentiated/dedifferentiated carcinoma or carcinosarcoma: CHT + EBRT/brachytherapy. * Unclear benefit of added CHT in stage I–II clear cell carcinomas
Advanced disease	Upfront surgery with tumor debulking if complete macroscopic resection is feasible with acceptable morbidity and QoL for the patient.	

**Table 3 cancers-15-01091-t003:** Management of recurrences.

Recurrence	ESGO-ESTRO-ESP, ESMO	NCCN
Local or regional	No prior radiotherapy exposure: EBRT + brachytherapy (1st choice) +/− CHT * Superficial vaginal cuff recurrences: brachytherapy alone * Consider surgery for solitary easily accessible superficial vaginal tumor prior to RT for better local symptom control.	No prior radiotherapy exposure: EBRT + brachytherapy (1st choice) +/− CHT
	Previous BRT only, surgical exploration: Disease confined to vagina or paravaginal soft tissues, EBRT with brachytherapy boost. If locoregional nodal disease, to pelvic or para-aortic lymph node, EBRT +/− CHT If upper abdominal or peritoneal recurrence, CHT + palliative RT if necessary.	Previous BRT only, surgical exploration: Disease confined to vagina or paravaginal soft tissue, EBRT plus brachytherapy If locoregional nodal disease, to pelvic or para-aortic lymph node, EBRT +/− CHT If upper abdominal or peritoneal recurrence, CHT + palliative RT if necessary.
	Previous RT at the recurrence site, surgical exploration with radical resection when feasible + CHT +/− RT. If surgery is not feasible, radical re-irradiation.	Previous RT at the recurrence site, surgical exploration with radical resection when feasible + CHT +/− RT. If surgery is not feasible, radical re-irradiation.
Isolated distant metastasis	Surgical resection if feasible (+/− CHT + RT) or selected stereotactic RT	Surgical resection if feasible or selected stereotactic RT
Disseminated metastasis/further recurrences	Low grade, asymptomatic, hormone receptor-positive metastases: hormonal therapy (CHT to progression) Symptomatic, high grade, large volume metastases: multiagent CHT (if tolerated) * carboplatin–paclitaxel first line. * consider single-agent options if indicated. No standard treatment for second-line therapy, but doxorubicin and paclitaxel are considered the most active therapies. MSI-H/dMMR tumors: pembrolizumab Platinum-based CHT re-challenge if relapse > 6 months since last platinum-based therapy Clinical trials or best supportive care are appropriate	Low grade, asymptomatic, hormone receptor-positive metastases: hormonal therapy (CHT to progression) Symptomatic, high grade, large volume metastases: multiagent CHT (if tolerated) * carboplatin–paclitaxel first line. No standard treatment for second-line therapy MSI-H/dMMR tumors: pembrolizumab (or nivolumab) Recurrent HER2 serous carcinoma: carboplatin/paclitaxel/trastuzumab bevacizumab or temsirolimus approved single-agent biologic therapy for progression on previous cytotoxic CHT. Clinical trials or best supportive care are appropriate.

**Table 4 cancers-15-01091-t004:** Follow-up scheme.

Follow-Up Scheme	ESMO	NCCN
Physical/gynecological examinations, TVUS	Low risk–every 6 months (consider phone f-up) for 2 years, then yearly. High risk–every 3 months for 3 years, then every 6 months up to 5th year, then yearly.	Every 3-6 months for 2 or 3 years, then every 6 months up to 5th year and then annually.
Serum CA 125	Not routinely recommended.	Only if initially elevated.
Pap smear	Not routinely recommended.	Not routinely recommended.
Routine CT scans	Only in high-risk group, every 6 months for the first 3 years and then on an individual basis.	Only in advanced disease at presentation, every 6 months for the first 3 years, every 6 to 12 months for 2 additional years.
PET CT/chest abdomen CT scan/MRI	In suspected cases.	In suspected cases.

## Data Availability

Not applicable.

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
