# Peer review of "Management of Patients Diagnosed with Endometrial Cancer: Comparison of Guidelines"

_cancers, 2023, doi:10.3390/cancers15041091_

Round 1

Reviewer 1 Report

I think this is an interesting way of drawing attention in a serious matter, although unifying guidelines is a sensitive goal.

The whole work is explained very clear and worth to be published.

Author Response

Thank you for your appreciated review,

kind regards.  

Reviewer 2 Report

I recommended the acceptance of this manuscript.

Author Response

I would like to thank you for your review, I appreciate the referrals. As concerns references, we found out many of them starting our research from the NCCN and ESGO-ESTRO-ESP guidelines. 

Reviewer 3 Report

I appreciated the article that I revised. It is about one of the main topics in gynecological cancers: endometrial neoplasia, new classification, new biological features, and treatment.

Authors compared main European and American guidelines and summarized differences in practical tables.

The writing is simple and linear, English sound ok with minimal spelling rephrasing.

It really allows you to move from theory to practice.

The paragraph relating to recurrence is also noteworthy.

Authors could not conclude what is the best one guideline but give clinician a useful tool.

All references are ok.

Author Response

Thank you for your recommendation. In the latest version I made a couple of minor changes and shortened some phrases. 

Kind regards